# From Mouth to Muscle: Exploring the Potential Relationship between the Oral Microbiome and Cancer-Related Cachexia

**DOI:** 10.3390/microorganisms10112291

**Published:** 2022-11-18

**Authors:** Shreya R. Raman, Christopher Liu, Kelly M. Herremans, Andrea N. Riner, Vignesh Vudatha, Devon C. Freudenberger, Kelley L. McKinley, Eric W. Triplett, Jose G. Trevino

**Affiliations:** 1Department of Surgery, Virginia Commonwealth University School of Medicine, Richmond, VA 23298, Canada; 2Department of Surgery, University of Florida College of Medicine, Gainesville, FL 32610, USA; 3Department of Microbiology and Cell Science, University of Florida, Gainesville, FL 32611, USA; 4Massey Cancer Center, Virginia Commonwealth University, Richmond, VA 23284, USA

**Keywords:** oral microbiota, inflammation, cytokines, microbiome, malignancy

## Abstract

Cancer cachexia is a multifactorial wasting syndrome associated with skeletal muscle and adipose tissue loss, as well as decreased appetite. It affects approximately half of all cancer patients and leads to a decrease in treatment efficacy, quality of life, and survival. The human microbiota has been implicated in the onset and propagation of cancer cachexia. Dysbiosis, or the imbalance of the microbial communities, may lead to chronic systemic inflammation and contribute to the clinical phenotype of cachexia. Though the relationship between the gut microbiome, inflammation, and cachexia has been previously studied, the oral microbiome remains largely unexplored. As the initial point of digestion, the oral microbiome plays an important role in regulating systemic health. Oral dysbiosis leads to the upregulation of pro-inflammatory cytokines and an imbalance in natural flora, which in turn may contribute to muscle wasting associated with cachexia. Reinstating this equilibrium with the use of prebiotics and probiotics has the potential to improve the quality of life for patients suffering from cancer-related cachexia.

## 1. Introduction

Only one of every two cells in the human body is human [1]. Of the remaining non-human cells, a significant portion is made up of bacteria and fungi. With each individual harboring 10 to 100 trillion microbial cells, these microorganisms play vital roles in maintaining a state of homeostasis [2]. The oral cavity contains the largest and most diverse collection of microorganisms in the body, second only to the gut microbiome [3,4]. Along with viruses and fungi, the oral microbiome is made up of over 700 different bacterial species that reside in the hard and soft palate, floor of the mouth, lips, tongue, teeth, gingiva, and the buccal mucosa [5]. At equilibrium, these microbiota function symbiotically with the human host. The oral microbiome contributes to metabolic, physiologic, and immunological functions that maintain the balance between human health and disease. These functions include digestion, nutrition, regulation of immune response, and prevention of disease-promoting microorganisms through maintenance of a mucosal barrier [6].

The microbial community of the oral cavity is complex and can be separated into core and variable microbiomes [4]. The core microbiome is composed of microflora that can be present throughout multiple organs in a person’s body when healthy [7]. This component of the oral microbiome is typically conserved across individuals [7]. For those in good states of health, the core microbiome should approximate the oral microbiome [8]. The concept of the core microbiome was explored in a study by Zaura et al. who capitalized on advances in sequencing techniques by performing 454 pyrosequencing of the oral microbiome in a group of healthy individuals. They found that even across several intra-oral microbial subcommunities including the dental surface, cheek, hard palate, tongue, and saliva, bacterial sequences of the oral microflora were for the most part homogenous [8]. Bik et al. also conducted a similar study in 2010 by amplifying bacterial sequences sourced from 26 distinct oral anatomic locations from 10 healthy individuals belonging to four discrete ethnicities [9]. They too, found identical bacterial sequences between their patients representative of the core microbiome. Specifically, they found evidence of the following genera’s presence in all their patients’ oral microbiomes: *Actinomyces*, *Atopobium*, *Corynebacterium*, *Rothia*, *Campylobacter*, *Cardiobacterium*, *Haemophilus*, *Neisseria*, *Tm7*, *Fusobacterium*, *Bergeyelia*, *Capnocytophaga*, *Prevotella*, *Granulicatella*, *Streptococcus*, and *Veillonella*. Notably, Bik et al. also found interindividual differences that were suggestive of the simultaneous presence of a distinct variable microbiome more specific to the individual.

The variable microbiome underlies the diversity in the oral microbiome seen between individuals [7]. This component is reflective of unique lifestyles, environments, and one’s own genotype [7].

The diet plays a prominent role in the variable oral microbiome composition [10]. Kato et al. explored the relationship between dietary composition and the oral microbiome through high-throughput 16S rRNA metagenomic sequencing. In their study, they identified correlations between alpha diversity indices (measuring variability between individuals) and saturated fatty acid [11]. Their sequencing analysis revealed that the abundance of *betaproteobacteria* and *fusobacteria* in the oral cavity was associated with dietary saturated fatty acid content. Additionally, they found that dietary glycemic load was positively correlated with *Lactobacillaceae* populations. A separate study examined the impact of vegan diets on the salivary microbiota. Broadly, Hansen et al. found significant differences between the oral microbiome of vegans and omnivores [12]. Analysis of patients’ primary dietary components showed that intake of fiber, medium chain fatty acids, piscine omega-11 mono-unsaturated fatty acids, and omega-3 polyunsaturated fatty acids was associated with salivary microbiota diversity. Dietary fiber was also associated with increased populations of *Capnocytophaga* and *Neisseria subflava* [12]. Together, these studies advocate for the recognition of another highly complex influence that the diet has on human health.

Ingestion of alcoholic beverages has also been postulated to have effects on the oral microbiome. Fan et al. sought to determine if differences in the oral flora existed between persons with different levels of alcohol use by performing 16S rRNA gene sequencing in 1044 American adults [13]. Between non-drinkers and heavy drinkers, the authors observed a significant difference in alpha and beta diversity. Furthermore, they noted significant differences in oral flora composition. *Actinomyces*, *Leptotrichia*, *Cardiobacterium*, and *Neisseria* were found to be greater in abundance in the heavy drinker group. Organisms belonging to the *Lactobacillales* order were found to be decreased in abundance when comparing heavy drinkers to non-drinkers. These differences persisted even after controlling for smoking status. Paralleling the findings of this study was a separate study of 150 healthy Chinese subjects comparing the microbiomes of those who drank alcohol and those who did not. Much like Fan et al.’s results, they also observed greater alpha diversity in alcohol drinkers and differences in the overall oral microbiome between drinkers and non-drinkers [14]. Their sequencing analysis uncovered that alcohol drinkers had greater organism populations of the *Prevotella* and *Moryella* genus as well as *Prevotella melaninogenica* and *Prevotella tannerae* species. On the other hand, *Lautropia*, *Haeophilus*, and *Porphyromonas* genera were diminished in the alcohol drinkers’ group. Decreased enrichment of *Haemophilus parainfluenza* populations was also observed. Interestingly, the enrichment of genera observed in the alcohol drinkers’ group was positively correlated with enhancement of anaerobic metabolic pathways and negatively correlated with the aerobic pyruvate metabolic pathway.

Cigarette smoking represents another environmental factor affecting the oral microbiome. A recent study of Jordanian subjects determined the existence of differences in the oral microbial communities of smokers and non-smokers. They specifically observed significant elevations in *Streptococcus*, *Prevotella*, and *Veillonella* genera and depression of *Neisseria* populations [15]. A similar study in the Chinese population reproduced the findings of enriched populations of *Actinomyces* and *Veillonella* in smokers [16]. The same experiment also found connections between cigarette use and *Moryella*, *Bulleidia*, and *Moraxella* genera as well as *Prevotella melaninogenica*, *Rothia dentocariosa*, *Prevotella pallens*, *Bulleidia moorei*, *Rothia aeria*, *Actinobacillus parahaemolyticus*, and *Haemophilus parainfluenzae* species. Further studies have added *Streptococcus sobrinus* and *Eubacterium brachy* as taxa also linked with positive smoking status [17]. These oral microbiome differences have led researchers to suggest the possible existence of a microbial signature able to differentiate smokers from non-smokers [15].

The increasing awareness of health disparities has led to the theorization of possible oral microbiome differences in persons of different socioeconomic statuses. A study conducted in a Danish cohort set out to test this hypothesis. To do this, Belstrøm et al. stratified patients into different socioeconomic statuses by scoring patients’ municipality of residence on various socioeconomic measures [17]. They then assessed saliva samples of their patients with a high throughput-based microarray platform. Bacterial probe assessment demonstrated significant differences in both presence and abundance of bacterial organisms between low and high socioeconomic status groups. *Veillonella parvula* taxa, *Veillonella atypica* taxa and *Streptococcus parasanguinis* clusters were highlighted to be different [17]. These effects of socioeconomic status can even be seen as early as five years of age. Boyce et al. studied dental health in kindergarten aged children of different upbringings residing in the San Francisco area to determine if a relationship between family socioeconomic status and oral microbiome composition existed [18]. Both socioeconomic status, measured by the parent-reported highest level of household education, and financial stress, measured by parental response to a 4-item questionnaire, were collected. These two variables were not surprisingly inversely related with each other. Importantly, socioeconomic status was associated with greater populations of cariogenic bacteria [18].These studies demonstrating differing microbiome composition in subjects of distinct socioeconomic classes offer a springboard for future investigations looking to elucidate the mechanisms of healthcare disparities.

Genetics are a non-modifiable factor known already to affect the gut microbiome with recent studies demonstrating similar effects on the oral microbiome as well. Characterization of monozygotic and dizygotic twin biofilm flora showed that oral microbiomes of monozygotic twins were more similar than those of dizygotic twins as measured by Bray–Curtis distances [19]. Highly hereditable oral flora identified in this study included *Prevotella pallens*, *Veillonella* taxon, *Pasteurellaceae*, *Corynebacterium durum*, *Leptotrihcia*, and *Abiotrophia*. A Colorado Twin Registry study produced similar results as they found greater beta diversity in dizygotic or unrelated individuals compared to monozygotic twins [20]. GWAS analysis was also performed and identified two loci, one located in proximity to the IMMPL2 gene on chromosome 7 and one near the INHBA0AS1 gene on chromosome 12, with the potential to determine oral microbiome phenotypes. Certainly, as demonstrated in these studies, understanding of both environment and genetic factors is critical in consideration of the variable microbiome.

When the balance of disease-preventing microbiota is tipped toward a disease-promoting microbial environment, previously inert bacteria may contribute to pathologic host response. Dysbiosis occurs when there is a shift in the composition or abundance of microbial communities deviating from homeostasis [21]. In this state, previously beneficial bacteria may lead to chronic and systemic inflammation in the body, resulting in disastrous health consequences. Oral dysbiosis has been implicated in numerous inflammatory diseases such as periodontitis, atherosclerosis, obesity, and cancer [22,23,24,25]. Although a direct mechanism has not been established between dysbiosis and the development of cancer, studies have continuously found that organisms from the oral cavity can influence a tumorigenic and inflammatory state [21,26]. This association has been studied extensively in malignancies of the abdominal cavity. In pancreatic cancer patients, increased populations of *Porphyromonas gingivalis, Aggregatibacter actinomycetemcomitans, Enterobacteriaceae, Lachnospiraceae G7, Bacteroidaceae, Granulicatella. adiacens, Leptotrichina, Streptococcus, and Staphylococcaceae* have been observed in the oral microbiome [27]. Colorectal cancer, another abdominal malignancy, has been linked to greater prevalence of *Peptostreptococcus, Parvimonas, and Fusobacterium* [28]. Even precancerous gastric cancer lesions have been demonstrated by Salazar et al. to be associated with *Porphyromonas gingivalis*, *Tannerella forsythia*, *Treponema denticola* and *Aggregatibacter actinomycetemcomitans* colonization of dental plaque [29].

Cancer cachexia is a clinical manifestation of the inflammatory host response to carcinogenesis. It is a multifactorial syndrome resulting in skeletal muscle and adipose wasting, as well as anorexia [5]. Cancer cachexia’s reach can even extend beyond the musculoskeletal system to involve the heart, leading to cardiac muscle wasting and ultimately heart failure [30,31]. It is estimated that 2 million people die annually as a consequence of cancer cachexia [32]. Half of all cancer patients will eventually develop cancer cachexia and 20% will die as a result of this syndrome [33]. Research into therapeutic options for cancer cachexia is ongoing, with no cure or effective treatment yet found [34]. Cachectic patients are often left fatigued as well as immunosuppressed. This state leaves patients unable to tolerate chemotherapy and renders them poor candidates for surgical resection, thus contributing to advancement of the cancer [35]. This vicious cycle eventually leads to a lower quality of life and overall survival [36,37].

The immunosuppression that cachectic patients experience is due to subdued nutritional intake as well as a heightened inflammatory state. Cancer cachexia has been directly associated with the cytokines involved in the inflammatory response. In a study by Riccardi et al., patients with cancer cachexia were found to have increased levels of circulating TNF-α, IL-6, and IL-8 when compared with non-cachectic cancer patients and controls [38]. CRP, IL-1, IFN-γ, and proteolysis inducing factor (PIF) have also been found to be increased in cachectic patients [39]. Activation of metabolic pathways by these aforementioned molecules then sets in motion the processes that directly underlie cancer cachexia [33]. The heightened inflammatory state present in cachectic patients is also mediated by reactive oxygen species. In cancer patients, elevated levels of reactive oxygen species are produced by increased uncoupling of the electron transport chain and resulting loss of mitochondrial membrane potential [40]. TNF-α driven upregulation of mitochondrial reactive oxygen species production as well as diminished antioxidant presence in muscle cells also contribute to the high levels of oxidative stress present in cancer patients [40]. Ensuing downstream upregulation of Ubiquitin Proteasome System activity, Calpain expression, and autophagic activation then proceed to induce muscle wasting [40,41,42,43].

Additionally, cancer cachexia has been associated with dysregulation of brain hypothalamic signals, leading to decreased appetite and metabolic changes which can contribute to lower quality of life [38]. Specifically, Braun et al., identified that the inflammatory marker IL-1 affects regulation of energy homeostasis in the central nervous system, leading to decreased food intake and anorexia [44]. These results indicate that central IL-1 production leads to malnutrition and increased resting energy expenditure that ultimately results in skeletal muscle catabolism [44]. Moreover, the resulting weight loss and fatigue from cancer cachexia devastates patient well-being. This relationship was highlighted in a secondary analysis of 405 cancer patients in a Swedish outpatient palliative care program [45]. In this analysis, Wallengren et al. found that weight loss above 2%, fatigue, BMI less than 20 kg/m^2^, and a CRP greater than 10 mg/L were significantly associated with adverse quality of life [45]. Weight loss above 2% and fatigue were additionally associated with shorter survival.

Currently, attempts to modify and increase nutritional intake are the primary treatment for cancer cachexia [36]. However, the use of nutritional supplements has not been shown to have clinical benefits of weight gain or improved function [46]. Palliative care and counseling is used secondarily to counteract weight loss and to reduce distress among the patient and their family. Pharmacologic therapies such as appetite stimulants, anabolic steroids, ghrelin, non-steroidal anti-inflammatory drugs, psychiatric drugs, and thalidomide are being studied to treat cachexia, although side effects are prevalent and effectiveness is limited [46]. Interventions involving physical activity have also been investigated as a potential avenue of therapy in cancer cachexia. Yet, no clear benefit has been determined. While one randomized controlled study in cachectic pancreatic cancer patients concluded benefits of resistance training in the aspects of muscle strength, mobility and lean body mass, results of a meta-analysis were unable to draw conclusions in regard to efficacy or safety [47]. Clinical studies with promising results and minimal side effects are rare but generate excitement. One such study by Maccio et al. treated cachectic signs and symptoms with a combination of megestrol acetate, EPA, L-carnitine, and thalidomide. This combination was shown to improve appetite and performance status without increased risk of toxicity when compared to patients taking monotherapies [48]. While these therapies continue to be explored further, the only true cure for cachexia is to cure the underlying disease; a task difficult in late stage or advanced solid tumor cancer patients.

Given the limited treatment options available for cancer cachexia, additional research is needed to elucidate potential therapeutic targets. While the relationship between the gut microbiome and inflammatory response has been previously investigated, research involving the oral microbiome remains sparse. This review highlights the importance of the oral microbiome in systemic inflammation and in cancer cachexia. It further explores potential interventions for patients suffering from cancer cachexia.

## 2. The Oral Microbiome and Inflammation

Inflammation is intricately involved in the growth and development of malignancies. In addition to the activation and recruitment of the innate and adaptive immune system, inflammation is critical for tissue repair and regeneration [49]. As cancer progresses, local and systemic inflammation become dysregulated, promoting both tumorigenesis as well as cancer-related cachexia. The inflammatory response consists of a combination of different immune mediators, with cytokines contributing to the phenotypic changes seen in tissues [50,51].

Oral dysbiosis is also implicated in systemic inflammation (Table 1). Changes in the oral microbiome may facilitate the development of systemic inflammation but also may be exacerbated by systemic inflammation, potentially leading to a vicious cycle (Figure 1).uIn a study by Sarkar et al., the oral microbiome was compared to the salivary levels of the inflammatory cytokines IL-1β, IL-6 and IL-8. Bacterial Operational Taxonomic Units (OTUs) belonging to *Prevotella*, SR1 and *Ruminococcaceae* were found to be associated with IL-1β whereas *Prevotella* and *Granulicatella* were associated with IL-8 [52]. *Prevotella*, in particular, is notable du eto its involvement in inflammatory diseases such as periodontitis, bacterial vaginosis, rheumatoid arthritis, and metabolic diseases [53]. The relationship between diurnal fluctuations in inflammatory markers and the oral microbiome was also explored in this study. Correlations in diurnal fluctuations were also observed with IL-1β and *Prevotella*, IL-6 and *Prevotella*, IL-6 and *Neisseria*, and IL-6 and *Porphyromonas*. Atarashi et al. separately demonstrated that oral bacteria were associated with the activation of a pro-inflammatory milieu by inoculating mice with the oral bacteria of patients with inflammatory bowel disease (IBD). Significant inflammation was observed via activation of the Th1 immune signaling pathway [54]. The impact of the oral microbiome on systemic inflammation may also be mediated by bacterial products. Bacterial extracellular vesicles carry a variety of substrates including microbe-associated molecular patterns and molecules sourced from the bacteria where they were generated. Depending on the specific bacteria that these vesicles come from, their contents can exert various effects on the host including the activation of immunostimulatory pathways [55,56]. As such, these microbial products may provide another mechanism by which oral dysbiosis causes systemic inflammation. Kim et al. examined this process by studying bacterial extracellular vesicles derived from periodontal pathogens and oral commensal bacterium [57]. They hypothesized that these bacterial products could influence the differentiation of osteoclasts, a cell type derived from the macrophage-monocyte cell lineage [58]. Extracellular vesicles from the periodontal pathogens *Porphyromonas gingivalis* and *Tannerella forsythia* as well as from the oral commensal bacterium *Streptococcus oralis* induced osteoclastogenesis through activation of Toll-like Receptor 2. *Porphyromonas gingivalis* uniquely has also been observed to produce a proinflammatory response through the same receptor pathway [59]. Indeed, these studies depict an interwoven relationship between dysbiosis and the onset of inflammation, a cornerstone to the beginning of systemic consequences.

The inverse of this cycle may also take place when systemic disease precipitates an inflammatory state. An example is seen in the relationship between diabetes and periodontitis. Approximately 60% of patients with Type 1 Diabetes have periodontitis compared to only 15% found in a control population of patients without Type 1 Diabetes [60,61]. The pathogenesis of this may be explained by the diabetes-induced secretion of pro-inflammatory cytokines such as IFN-γ, TNF-α, and IL-6 [62,63]. This cytokine production triggers a systemic innate immune response that involves the oral cavity. Over time, this can alter the natural flora of the oral microbiome, resulting in dysbiosis.

Systemic lupus erythematosus (SLE) has similarly been examined in relation to inflammatory response and its effect on the oral microbiome. Jensen et al. first established this in the 1990 s by comparing bacterial loads in SLE and healthy subjects, ultimately finding that SLE patients had higher bacterial loads [64]. A study by Corrêa et al. built on these findings by sampling subgingival dental plaques of 52 SLE and 52 control patients to interrogate the relationship between SLE and subgingival bacterial community composition. They found that *Prevotella nigrescens*, *Prevotella oulorum*, *Prevotella oris* and *Selenomomonas noxia* species were enriched in healthy periodontal sites of SLE patients compared to non-SLE patients. At periodontitis sites, SLE patients exhibited significantly greater abundance of *P.ouloorum*, *Fretibacterium fastidiosum* and *Anaeroglobus germinatusin* in addition to *Fusobacterium* taxas 360 450 and *TM7* taxon 437 [65].

Associations between the oral microbiome and mental health were recently discovered [66]. Using saliva samples obtained from students at the University of Florida, Ahrens et al. sought to establish a connection between salivary microbiota and recent suicidal ideation. Here, *Alloprevotella rava* was found in significantly higher relative abundance in those students with no suicidal ideation, particularly in the absence of the minor allele “G” at SNP rs10437629. *Alloprevotella rava* ferments glucose to succinate which is known to improve glucose oxidation in the brain and brain metabolism after injury [67,68,69,70,71].

The systemic inflammation of cancer cachexia is a result of similar mechanisms. Upregulation IL-6, IL-8, TNF-α and IFN-γ may occur as a response to homeostatic shifts in the oral microbiota and lead to further amplification of inflammatory cytokines throughout the body. Resultant chronic inflammatory states may potentially stimulate cell proliferation and contribute to tumorigenesis [72]. In turn, tumor growth and metastasis leads to further increased inflammation (Figure 1). The systemic inflammatory state may then exacerbate oral immune dysregulation, potentially rendering the oral microbiome susceptible to pathogenic invasion. As found in diabetes and SLE, this dysbiosis has the potential to perpetuate an ongoing cycle of inflammation, worsening cancer cachexia. Though it may be difficult to distinguish the sequence of events, it is feasible that the reduction of inflammation may reduce dysbiosis.

## 3. The Microbiome and Cancer Cachexia

The relationship between the oral microbiome and cancer cachexia is a novel field of research that remains unexplored. However, the gut microbiome in the distal alimentary tract has been implicated in cancer cachexia [5]. Gut barrier dysfunction as well as an imbalance in the gut microbiome has been shown to lead to systemic inflammation, setting the stage for a cachectic response in cancer patients [5]. Jiang et al., specifically examined the role of bacterial translocation in the colon of patients with cancer cachexia. In this study, cachectic patients had a significantly higher prevalence of colonic bacterial translocation when compared to non-cachectic patients. Cachectic patients also exhibited increased concentrations of IL-6, TNF-α, and IFN-γ in their venous blood samples from the middle colic vein. The study concluded that luminal bacterial overgrowth likely occurs secondary to immunosuppression, resulting in increased bacterial translocation. This induces a pro-inflammatory state, raising the metabolic rate and suppressing appetite. This cycle may result in further bacterial translocation which will then lead to a greater cytokine release eventually resulting in a cachectic state, a cycle similar to the one that may potentially be occurring in the oral microbiome [73].

The gut microbiome and cachexia were further explored in a 2018 study by Bindels et al. who investigated markers of gut barrier function by assessing mice injected with colon adenocarcinoma cells [74]. In these subjects, intestinal morphology was altered, renewal of cell lineages was decreased, and a decreased expression of the tight junctions responsible for binding the epithelium was observed [74]. These findings are supported by similar studies in leukemic mice with cachexia and a colorectal cancer mouse model of cachexia [75,76]. Both investigations indicated that gut barrier dysfunction was implicated in cachexia through its induction of a systemic inflammatory state. On a similar note, intestinal morphology is also altered by the binding of gut microbiota to gut epithelium with cadherin junctions, a type of adhesion molecule [77]. One such bacteria that illustrates this concept is *Porphyromonas gingivalis*, a bacterium that degrades cadherin junctions, leading to the translocation of *Porphyromonas gingivalis* and a state of systemic inflammation [78].

More knowledge of the relationship between the gut microbiome and cancer cachexia may be gleaned from interventional studies. Sakakida et al. studied the effects of partially hydrolyzed guar gum (PHGG), a soluble dietary fiber, in a preclinical experiment [79]. They hypothesized that PHGG’s effect on the intestinal flora may counteract the pro-inflammatory intestinal state that leads to cancer cachexia. Utilizing a colon-26 murine cachexia model, they found that non-PHGG fed mice had decreased skeletal muscle mass compared to those with diets containing PHGG. Mice fed with PHGG also demonstrated increased *Bifidobacterium*, *Akkermansia*, and an unspecified *S24-7 family* populations with an associated preservation of gut barrier function. The resulting decrease in systemic levels of pro-inflammatory lipopolysaccharide-binding protein and IL-6 substantiates Jiang et al.’s, proposed role of gut permeability and bacterial translocation in the pathogenesis of cancer cachexia [74]. Interestingly, obesity and diabetes have been postulated to have similar elements in their pathology [80]. A separate investigation by Jia et al. has recently applied the anti-inflammatory effects of eggshell membranes observed in joint and connective tissue preservation to the field of cancer cachexia [81]. Its low-cost and its recognized ability to remedy intestinal dysbiosis make this modality an especially promising therapeutic candidate [81]. This multi-component investigation of eggshell membrane effects in an IL-10-knockout murine model of cachexia yielded a variety of key results. In particular, they found that the gut microbial make-up of mice receiving eggshell membranes differed from those who received non-supplemented diets. *Bacteriodetes*, *Firmicutes*, and *Verrucomicrobia phyla*, *Bacteroidacae*, *Defferribacteraceae*, *Ruminococcaceae*, and *Porphyromonadacae* families, and *Bacteroides ovatus*, *Bacteroides acidifaciens*, and *Akkermansia Muciniphila* species were among the microbiota whose population sizes increased to wild-type levels with eggshell membrane supplementation [82]. The restoration of *Ruminococcaceae* populations was of particular interest as this organism is known to have the potential to stimulate fermentation of short chain fatty acids which can induce wide-spread anti-inflammatory effects [82].

With regard to function and regulation, the oral and gut microbiome share many similarities. Both play a role in immune defense and house some of the largest stores of natural flora in the body [83,84]. The oral microbiome also serves as the introductory site for gut dysbiosis [84]. Inflammatory signals via cytokine signaling caused by bacteria in the oral cavity may be secreted down the digestive tract and influence the gut microbiome [85]. Additionally, oral dysbiosis has the potential to lead to foreign bacteria translocation to the gut, causing chronic and systemic inflammation [86]. While further studies are necessary, parallels between both cavities may indicate that, similar to the gut, the oral microbiome plays a significant role in the onset and perpetuation of inflammation leading to cancer cachexia.

## 4. Potential Therapeutic Agents

Biotherapeutics such as prebiotics and probiotics may be effective tools that have the potential to restore the balance of the oral microbiome. By re-establishing equilibrium among certain bacteria of the oral cavity, clinicians may effectively reduce the extent of muscle wasting seen in cancer cachexia.

Probiotics are defined as “viable micro-organisms that provide health benefits when taken in sufficient doses” [87]. At its essence, they are live bacteria used to displace pathogenic bacteria. Studies on the effects of probiotic administration on cancer cachexia through alteration of the oral microbiome are limited. However, the literature that exists regarding their effects on the oral cavity itself provides insight into their ability to impact local inflammation and perhaps oral microflora composition. Administration of probiotics in periodontal disease has been shown to decrease pro-inflammatory cytokines in gingival crevicular fluid and myeloperoxidase activity [88,89]. Additional studies of their effects in dental caries and dental plaque have supported their ability to resolve these oral cavity diseases through the reduction of *streptococcus mutans* populations [90,91,92,93]. Similarly, growth of gingivalis and halitosis associated bacteria has been observed to be hampered through the inhibitory effects of probiotics on *Porphyromonas gingivalis* [94,95,96]. Even voice prosthetics have been found to have an increased lifespan due to probiotic use because of probiotics’ prohibiting effect on microorganism adhesion [97]. Mechanistically, these probiotic effects are made possible through competition for adhesion, production of anti-microbial compounds, and enhancement of host immune response [98].

Preclinical studies assessing the effects of probiotics on colorectal cancerous and precancerous lesions have also suggested an anticancer role. One such experiment reported that *Pediococcus pentosaceus FP3*, *Lactobacillus salivarius FP25*, L. *salivarius FP35*, and *Enterococcus faecium FP51* inhibited cancer proliferation in an in vitro model of colorectal cancer [99]. The authors though believed that these anticancer properties were due to production of short chain fatty acids and did not note the alteration of the oral microbiome as an involved mechanism. However, another study conducted by Radaic et al. did indicate that probiotic effects in the oral microbiome served as the driver of anticancer activity [100]. Their investigation described the ability of the *Lactococcus lactis* probiotic to produce nisin, an antimicrobial substance active against Gram-positive and negative organisms [100]. Their findings in combination with nisin’s ability to increase apoptosis and survival in head and neck cancer models suggests an anticancer probiotic property mediated by oral microbiome modification [101].

Better studied are the effects of probiotics on the gut microbiome and its subsequent effects on systemic inflammation. Probiotics have previously been shown in studies analyzing intestinal inflammation from Crohn’s disease to have anti-inflammatory effects by decreasing pro-inflammatory cytokines such as IL-8 and IL-1β [102]. Probiotic treatment can also maintain gut flora, improve flora defensive capabilities against pathogenic colonization, and improve intestinal barrier integrity [103]. One such study has been able to link these systemic effects to age related cachexia, a disease process with mechanisms that overlap with those of cancer cachexia. Chen et al. investigated the use of the probiotic *Lactobacillus casei* Shirota in SAMP8 mice models of age-related cachexia. They found that supplementation of *Lactobacillus casei* Shirota was associated with decreased inflammation, reduced age-related increases in reactive oxygen species, and altered gut microbiota composition. Importantly, key findings also included attenuated declines in muscle mass, strength, and mitochondrial function within the *Lactobacillus casei* Shirota supplemented group [104].

Prebiotics are a “non-viable food component that confer health benefits on the host associated with modulation of the microbiota” [105]. These substances act as nutritional sources for healthy bacteria and allow them to expand their population in the oral cavity. Recent studies have indicated applications of their utilization in the promotion of healthy oral flora. Slomka et al. studied the effects of 742 compounds on the respiratory activity in 16 oral bacteria [106]. Out of all the compounds studied, they identified beta-methyl-d-galactoside and N-acetyl-d-mannosamine as prebiotics with the ability to stimulate the growth of beneficial bacteria in the oral microbiome [106]. Rosier et al. advocated for the classification of nitrate as a prebiotic after observing that its administration in in vitro biofilms increased levels of beneficial genera *Neisseria* and *Rothia* and decreased populations of the dental disease associated genera *Streptococcus*, *Veillonella*, *Porphyromonas*, *Fusobacterium*, *Leptotrichia*, *Prevotella*, *Alloprevotella* and *Oribacterium* [107]. Though their ability to affect cancer cachexia through the oral microbiome has never been established, prebiotic therapy has been shown to have beneficial effects in models of cancer cachexia via effects on the gut microbiome. Triterpine saponins and pectic oligosaccharides specifically are prebiotics named as being potentially therapeutic. In 2017, Huang et al. reported results of a study assessing the effects of ginsenoside-Rb3 and ginsenoside-Rd, specific types of triterpene saponins, in a mouse model of colorectal cancer [108]. Their analysis demonstrated that both triterpene saponins exerted anti-inflammatory effects on the mucosal cytokine profile. Strikingly, *Dysgonomonas* and *Helicobacter*, species of cancer cachexia associated bacteria, were decreased in prebiotic treated mice [108]. In a study with cachexia specific endpoints, Bindels et al. studied both inulin and pectic oligosaccharide prebiotics in the synbiotic treatment of a murine leukemia model. While inulin supplementation decreased leukemic invasion of the liver, pectic oligosaccharides were associated with increased *Bifidobacterium*, *Roseburia* and *Bacteroides* species in the gut. Most importantly, pectic oligosaccharide prebiotics delayed onset of cancer cachexia and decreased fat mass loss via modulation of genes involved in ß-oxidation [75].

Studies exploring interventions that alter the oral microbiome to potentially treat cancer cachexia are still in fledgling stages (Table 2). Agents that can re-establish and maintain oral symbiosis may also have properties to address the chronic and systemic inflammation associated with cancer cachexia.

## 5. Conclusions

The oral microbiota is a vast and diverse collection of bacteria that plays an important role in maintaining health. Composition of this microbial environment is complex and is determined by a variety of internal and external factors. Direct links between oral microbiota and inflammatory diseases remain largely unexplored, offering an exciting field of future study. Cancer cachexia is a devastating consequence of the systemic inflammation from tumorigenesis. While no therapeutic agents currently exist to treat this multifactorial syndrome, exploration of interventions targeting the gut microbiome have shown promise. Both prebiotic and probiotic agents are among the therapeutic modalities currently under investigation. Their reported ability to mitigate cancer cachexia through alterations in the gut microbiome suggests a role for not only gut flora but also oral flora as targets for future treatment. The effects of oral microbiome intervention and cancer cachexia, however, have yet to be explored and further study is needed to interrogate this potential relationship.

## Figures and Tables

**Figure 1 microorganisms-10-02291-f001:**
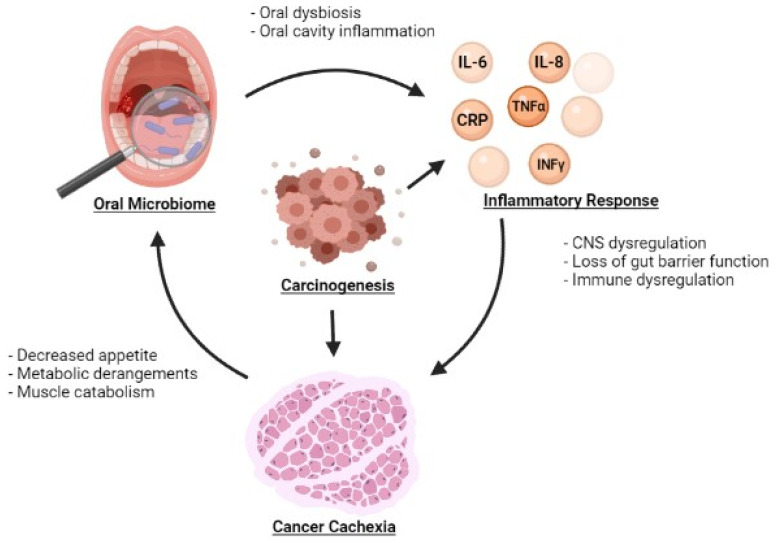
Relationship between the oral microbiome, inflammatory markers, and development of cancer cachexia.

**Table 1 microorganisms-10-02291-t001:** The Oral Microbiome and Inflammation.

Author (Year)	Study Participants (n)	Findings	Location
Clinical Studies			
Sarkar et al., 2021 [52]	12 healthy human subjects	*Prevotella*, SR1, and *Ruminococcaceae* are associated with IL-1β *Prevotella* and *Granulicatella* are associated with IL-8. Connections exist between IL-1β and *Prevotella* in regard to periodicity. As well as between IL-6 and *Prevotella*, *Neisseria*, and *Porphyromonas*	United States
Poplawska-Kita et al., 2014 [60]	107 Diabetic Patients and 40 Healthy Controls	Type 1 Diabetes increases risk of periodontal disease. Patients with periodontitis had higher levels of TNF- α	Poland
Jensen et al., 1999 [64]	93 SLE patients	Bacterial oral microbiome loads in SLE patients were greater than those of healthy subjects	Norway
Corrêa et al., 2017 [65]	52 SLE and 52 control patients	In healthy periodontal sites, *Prevotella nigrescens*, *Prevotella oulorum*, *Prevotella oris*, and *Selenomomnas noxia* populations were increased in SLE patients. In periodontitis sites, SLE patients had greater populations of *P. ouloorum*, *Fretibacterium fastidiosum* and *Anaeroglobus germinatusin*.	Brazil
Ahrens et al., 2022 [66]	489 Undergraduate and Graduate Students at the University of Florida	*Alloprevotella rava* was in greater abundance in students with no suicidal ideation, particularly those who do not have the minor “G” allele at SNP rs10437629.	United States
Pre-clinical Studies			
Atarashi et al., 2017 [54]	Mice transplanted with the saliva of patients with IBD	Oral bacteria can be associated with activation of a pro-inflammatory milieu. Inflammation was driven by activation of the Th1 immune signaling pathway	N/A
Kim et al., 2022 [57]	Bacterial extracellular vesicles of *Porphyromonas gingivalis*, *Tannerella forsythia*, *Streptococcus oralis*, and *Lactobacillus reuteri*	Extracellular vesicles from *Porphyromonas gingivalis*, *Tannerella forsythia*, and *Streptococcus oralis* induced osteoclastogenesis through activation of Toll-like receptor 2	N/A

**Table 2 microorganisms-10-02291-t002:** The Microbiome and Cancer Cachexia.

Author (Year)	Study Participants (n)	Findings	Location
Clinical Studies			
Jiang et al., 2014 [74]	1753 gastric cancer patients	Colonic bacterial translocation was significantly elevated in cachectic patients compared to non-cachectic patients. IL-6, TNF-α, and IFN-γ were also increased in cachectic patients.	China
Twetman et al., 2009 [88]	42 healthy adults with gingival inflammation	Patients receiving probiotic chewing gum had significant decreased levels of pro-inflammatory cytokines	Denmark
Staab et al., 2009 [89]	50 healthy adults	Ingestion of a probiotic drink once a day reduced myeloperoxidase and elastase activity in gingival crevicular fluid samples	Germany
Srivastava et al., 2016 [90]	60 Dental Cavity-Free Adults	Ingestion of probiotic curds was associated with reduction of *Streptococcus mutans* populations in saliva samples.	India
Wattanarat et al., 2015 [91]	60 School aged children	Probiotic supplementation was associated with reduction in populations of *Streptococcus mutans*.	Thailand
Nishihara et al., 2014 [92]	64 healthy adults	Probiotic administration of *L salivarius WB21* was associated with decreased levels of *Streptococcus mutans*	Japan
Chuang et al., 2011 [93]	80 healthy adults	Patients treated with *L. paracasei GMNL-33* containing probiotics had reduced levels of *Streptococcus mutans* compared to those treated with placebo tablets	Taiwan
Pre-clinical Studies			
Bindels et al., 2018 [73]	C26 Colon Carcinoma Mouse Cachexia Model	Alterations in gut permeability, epithelial turnover, gut immunity and microbial dysbiosis were observed.	N/A
Bindels et al., 2016 [75]	BaF Leukemic Mouse Model	*Lactobacillus* levels were decreased and *Enterobacteriaceae* levels were increased in the gut microbiome of cachectic mice. Restoration of these bacterial levels led to restored intestinal gut barrier function, decreased inflammation levels, reduced cancer-burden, and improved cancer-related cachexia.. Inulin supplementation decreased leukemic invasion of the liver, increased *Bifidobacterium*, *Roseburia*, and *Bacteroides* gut species were associated with pectic oligosaccharides. Peptic oligosaccharide administration was associated with delayed cancer cachexia and decreased fat mass loss.	N/A
Sakakida et al., 2022 [79]	C26 Colon Carcinoma Murine Cachexia Model	Partially hydrolyzed guar gum (PHGG) fed mice had increased skeletal muscle mass, preservation of gut barrier function, and decreased levels of of lipopolysaccharide-binding protein and IL-6. Levels of *Bifidobacterium*, *Akkermansia*, and an unspecified *S24-7 family* were associated with PHGG administration.	N/A
Jia et al., 2017 [82]	IL-10 knockout Murine Cachexia Model	Diet supplementation with eggshell membranes was associated with enrichment of Bacteriodetes, Firmicutes, and Verrucomcrobia phyla, Bacteroidacae, Defferribacteraceae, Ruminococcaceae, and *Poprhyromonadacea familes*, and *Bacteroides ovatus*, *Bacteroides acidifaciens*, and *Akkermansia Muciniphila* species.	N/A
Lee et al., 2014 [94]	in vitro	Mixing spent culture medium of *Streptococcus thermophilus* with spent culture medium of *P. gingivalis,* led to decreased levels of *P. gingivalis*	N/A
Suzuki et al., 2016 [95]	in vitro	*E. faecium WB2000* decreased *P. gingivalis* levels after co-culture.	N/A
Khalaf et al., 2016 [96]	in vitro	Inhibition of *P. gingivalis* growth was associated with *Lactobacillus* and *bacterioicin* from *L. plantarum*.	N/A
Schwandt et al., 2005 [97]	in vitro	Yakult Light fermented milk extended tracheoesophageal voice prostheses by a factor of 3.76	N/A
Thirabunyanon et al., 2013 [99]	in vitro	Probiotic lactic acid bacteria derived from infant feces inhibited cancer proliferation of colon cancer cells	N/A
Radaic et al., 2020 [100]	Biofilm derived from saliva samples of 10 healthy volunteers	Nisin producing *L. lactis* probiotic reduces oral biofilm formation.	United States
Chen et al., 2022 [104]	SAMP8 murine age-related cachexia model	Supplementation of *Lactobacillus casei* Shirota was associated with decreased inflammation, levesl of reactive oxygen species and alteration of gut microbiota. This treatment was also associated with attenuated declines in muscle mass, strength, and mitochondrial function.	N/A
Somka et al., 2017 [106]	in vitro	Beta-methyl-d-galactoside and N-acetyl-d-mannosamine prebiotics stimulated growth of beneficial bacterial microflora.	N/A
Rosier et al., 2020 [107]	in vitro	Nitrate was associated with increased levels of beneficial genera *Neisseria* and *Rothia* as well as dental disease associated genera *Streptococcus*, *Veillonella*, *Porphyromonas*, *Fusobacterium*, *Leptotrichia*, *Prevotela*, *Alloprevotella*, and *Oribacterium.*	N/A
Huang et al., 2017 [108]	APC^Min/+^ colorectal cancer mouse model	The triterpene saponin prebiotics, ginsenoside-Rb3 and ginsenoside-Rd, were observed to have anti-inflammatory effects on the mucosal cytokine profile. Cancer cachexia associated bacteria Cancer cachexia associated bacteria *Dysgonomonas* and *Helicobacter* were also decreased in mice who received prebiotic treatment.	N/A

## Data Availability

Not applicable.

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
