# Peer review of "From Mouth to Muscle: Exploring the Potential Relationship between the Oral Microbiome and Cancer-Related Cachexia"

_microorganisms, 2022, doi:10.3390/microorganisms10112291_

Round 1
Reviewer 1 Report
I am honored to examine a review article titled "From the Mouth to Muscle: A Review on the Oral Microbiome and Cancer-related Cachexia" by Raman et al., for publication in Microorganisms.
Introduction
"The rest are microbiota, made up of bacteria, viruses, and fungi that inhabit multiple sites throughout the human body" seems to suggest that viruses are also cells. This should be corrected. The next sentence assumes also that viruses are microorganisms, which is untrue.
Personnally I would leave visures out of these stats and focus on bacteria and fungi.
Please make sure to add references to all important claims
The section is too long and should be improved for concision
The Oral Microbiome and Inflammation
Please make sure to add references to all important claims
"Downstream effects of these molecules are particularly im-portant due to their ability to activate the NF-κB pathway". This is confusing. In fact IL-6, IL-1b and TNF-a are induced by NF-kb signaling in many inflammatory settings.
Others
I would have a separate paragraph for probiotics
The authors should provide clear perspectives for the field. " Currently, no therapeutic agents exist to treat this multifactorial syn-drome, leading to decreased overall survival following a cancer diagnosis. Promising re- sults of gut microbiome intervention in the treatment of cancer cachexia suggests a poten- tial role for therapeutics that target the oral microbiome" is not enough and should be substantially expanded.
Author Response
Reviewer 1
"The rest are microbiota, made up of bacteria, viruses, and fungi that inhabit multiple sites throughout the human body" seems to suggest that viruses are also cells. This should be corrected. The next sentence assumes also that viruses are microorganisms, which is untrue.
Personnally I would leave visures out of these stats and focus on bacteria and fungi.
We thank Reviewer 1 for their attention to detail and have adjusted the sentence to only discuss fungi and bacteria in order to reflect this important distinction as suggested.
Please make sure to add references to all important claims
We thank Reviewer 1 for their dedication to scientific writing etiquette and have added references to all important claims.
The section is too long and should be improved for concision
The Oral Microbiome and Inflammation
We thank Reviewer 1 for their regard to concise scientific writing. We have pared down the section entitled “The Oral Microbiome and Inflammation” by approximately 20% as suggested.
"Downstream effects of these molecules are particularly im-portant due to their ability to activate the NF-κB pathway". This is confusing. In fact IL-6, IL-1b and TNF-a are induced by NF-kb signaling in many inflammatory settings.
We thank Reviewer 1 for their expertise and have removed this sentence entirely in order to eliminate confusion and promote the section’s concision.
Others
I would have a separate paragraph for probiotics
We thank Reviewer 1 for their recommendation on content organization and have now included a section entitled “Potential Therapeutic Agents” to further structure the manuscript.
The authors should provide clear perspectives for the field. " Currently, no therapeutic agents exist to treat this multifactorial syn-drome, leading to decreased overall survival following a cancer diagnosis. Promising re- sults of gut microbiome intervention in the treatment of cancer cachexia suggests a poten- tial role for therapeutics that target the oral microbiome" is not enough and should be substantially expanded.
We thank Reviewer 1 for their recommendation and have edited this section for clarity.
Reviewer 2 Report
The authors noted that oral microbiome was associated with systemic inflammation and also had a certain effect on the nervous system, which was similar to inflammation and central disorders in cachexia. The authors speculated that the oral microbiome could play a role in cachexia, just like gut microbiome did. Oral microbiome could be disturbed by cachexia and in turn aggravate inflammation and anorexia. Intervention methods developed on this basis may become a new research direction.
The review is clear and innovative. However, the relationship of oral microbiome and cachexia is not that clear and there’s a lack of direct evidence, so I suggest modify the title.
Author Response
Reviewer 2
The review is clear and innovative. However, the relationship of oral microbiome and cachexia is not that clear and there’s a lack of direct evidence, so I suggest modify the title.
We thank Reviewer 2 for their time and feedback on the appropriateness of this paper’s title. The title has been revised to “From Mouth to Muscle: Exploring the potential relationship between the Oral Microbiome and Cancer-Related Cachexia”
Round 2
Reviewer 1 Report
My comments have been addressed